# Drought Induced Dynamic Traits of Soil Water and Inorganic Carbon in Different Karst Habitats

Liang Luo [1,2], Yanyou Wu [1,*], Haitao Li [3], Deke Xing [4,*], Ying Zhou [1,2] and Antong Xia [1,2]

1 State Key Laboratory of Environmental Geochemistry, Institute of Geochemistry, Chinese Academy of Sciences, Guiyang 550081, China
2 Institute of Geochemistry, University of Chinese Academy of Sciences, Beijing 100049, China
3 Department of Agricultural Engineering, Guizhou Vocational College of Agriculture, Qingzhen 551400, China
4 Key Laboratory of Modern Agricultural Equipment and Technology, Ministry of Education, College of Agricultural Engineering, Jiangsu University, Zhenjiang 212013, China
* Correspondence: wuyanyou@mail.gyig.ac.cn (Y.W.); xingdeke@ujs.edu.cn (D.X.)

**Abstract:** Understanding the temporal variability of soil water and carbon is an important prerequisite for restoring the vegetation in fragile karst ecosystems. A systematic study of soil moisture and carbon storage capacity under drought conditions in different karst habitats is critical for cultivating suitable crops in karst regions. The hydrological characteristics of soil and changes in soil $HCO_3^-$, pH, and EC values under drought conditions were measured on simulated rock outcrops and non-outcrops in an indoor pot experiment. The results showed that the rock outcrops had less evaporation and significantly greater water retention capacity than the non-outcrops, which gave the retained water in the rock outcrops sufficient reaction time to dissolve atmospheric $CO_2$, as well as to promote dissolution at the rock–soil interface. Therefore, the carbon sequestration capacity of the rock outcrops was higher than that of the non-outcrops. Due to the rock–soil–water interaction in the early stage of drought, the soil $HCO_3^-$ concentration in the rock outcrops fluctuated with soil water content, but the soil $HCO_3^-$ concentration tended to be stable in the whole drought period, showing a phenomenon of zero-carbon sink. No obvious change was observed in the soil $HCO_3^-$ concentration in non-outcrops during the drought period, which indicated that the carbon sequestration of rock outcrops was mainly attributed to the dissolution of rocks. Therefore, rock outcrops were more effective for water and carbon storage, compared with non-outcrops, under drought, and could provide more available water and carbon resources for supporting the photosynthesis of plants in karst regions.

**Keywords:** water retention function; carbon sequestration; carbon sink; rock outcrops; drought

## 1. Introduction

Karst ecosystems, in which exposed rocks and shallow soils form a mosaical pattern, change the distribution and transport of the surrounding soil water directly or indirectly, resulting in a high degree of spatial and temporal heterogeneity in the water distribution in karst regions [1]. The emergence of rock outcrops is very common in terrestrial ecosystems. Compared with the continuous and flat soils in non-outcrop habitats, the appearance of rock outcrops destroys the continuity of soil layers in karst areas, blocks the interconnection between soil patches, and causes the soil properties in this area to show a high degree of spatial variation [2]. At present, most studies focus on the dynamic changes in soil water and nutrients, and plant growth in outcrop habitats. Zhao et al. [3] found that surface outcrops altered rainfall runoff processes, intercepting and collecting large proportions of rainwater delivered to nearby soil patches, which significantly altered the permeability and water flow behavior of karst soils. This phenomenon is known as the "funnel effect" of exposed stones [4]. Rock outcrops play an important role in hydrology and ecology in drought areas. Certini et al. and Poesen et al. discovered that the rock fragments not only altered soil erosion processes and runoff generation, but also affected soil hydrological

processes by modifying soil physical and chemical properties [5,6]. Furthermore, Li et al. discovered that the more advanced the karstification process was, the greater its water catchment capacity [7]. The presence of rock outcrops modifies the microenvironment. Shen et al. [8] discovered that, in a soil and rock outcrop combination of different karst ecosystems, soil in rock outcrops obtains water, TOC, N, P, and K through rock surface runoff. Outcrops redistribute the water and nutrients received from the rocks to nearby soil patches, creating a specific rock outcrop–soil patch system [8,9], and releasing $Ca^{2+}$, $Mg^{2+}$, and $HCO_3^-$, causing a heterogeneous distribution of environmental factors such as soil water, inorganic carbon, and mineral nutrients in karst areas. Rock outcrops not only affect the redistribution of resources involving light, temperature, and water, but also influence the local climate and vegetation [10], causing spatial and temporal heterogeneity in the soil's hydrological characteristics and creating a range of diverse plant ecological niches [11]. The vegetation community composition is strongly influenced by the water content of soils. Furthermore, the heterogeneous distribution of inorganic carbon in karst areas is also influenced by soil water, which affects the local soil physicochemical properties and even vegetation growth. The semi-arid and semi-humid environmental conditions in karst areas hinder the progress of vegetation and ecosystem restoration. Therefore, understanding the water holding capacity of soils around rock outcrops under continuous drought conditions is necessary.

Water in the soil is highly efficient in the corrosion of carbonate rocks; when the partial pressure of $CO_2$ and water in the soil solution increases, the dissolution of carbonate rock can be promoted and the karst carbon sink effect can be strengthened [12]. On the other hand, low water contents mobilize and precipitate low amounts of calcium carbonate [13]. One mole of $CO_2$ is returned to the atmosphere for each dissolved mole of atmospheric $CO_2$ in carbonate, which is then precipitated as loam-forming carbonate [14]. During karstification, carbonate rocks have an effect on the atmospheric $CO_2$ source–sink relationship, both as a sink (carbonate dissolution) and as a source ($CO_2$ degassing in carbonate deposition) [15,16]. The specific process is as follows [17]:

$$H_2O_{(l)} + CO_{2(g)} \leftrightarrow H_2CO_3 \leftrightarrow HCO_3^- + H^+ \tag{1}$$

$$CaCO_3 + H_2O + CO_2 \leftrightarrow Ca^{2+} + 2HCO_3^- \tag{2}$$

In the above-mentioned equation, the carbonic acid solution formed by the combination of soil $CO_2$ and water is rapidly decomposed into $H^+$ and $HCO_3^-$ [18]. Calcium carbonate ($CaCO_3$) is influenced by soil $CO_2$, soil pH, soil water content, and soil calcium concentration [19]. The pH of the soil solution is influenced by the partial pressure of $CO_2$ originating from ecosystem respiration which, in turn, is controlled by the temperature and water availability [20]. Meanwhile, carbonates are alkaline salts which release a mass of $HCO_3^-$ to the soil and increase the pH of the soil solution. As a result, the increasing pH creates an adverse effect for plants that adapt to acidic environments, but suitable conditions for plants that adapt to basic soil conditions [21]. High concentrations of $HCO_3^-$ in soil solutions can affect plant growth [22] and inhibit plant photosynthesis [23], while low concentrations of $HCO_3^-$ can induce plant stomatal closure [24]. Consequently, monitoring the concentration of soil $HCO_3^-$ in different habitats can serve as a reference for cultivating adapted crops.

The dissolution processes of carbonate rocks is closely related to water and is easily affected by drought [25]. Some studies have shown that climate affects the partial pressure of $CO_2$ within karst systems. In the case of calcite, near equilibrium carbonate groundwater under warmer climatic conditions shows higher $pCO_2$ than that under cooler climatic conditions [26,27].Droughts are currently an important stressor for the carbon sink function of terrestrial ecosystems, which can significantly reduce the intensity of terrestrial ecosystem carbon sinks and even turn them into carbon sources [28]. It has been found that semi-arid ecosystems can become a large carbon sink in a wet year [29], fluctuating with changes in the external environmental conditions, and this fluctuation is closely related to changes in the

environmental conditions such as climate [30]. During the dissolution of carbonate rocks, carbonates might not capture any additional $CO_2$; instead, they will be reactivated from one pool to another and balanced with the atmosphere over time [31–33]. Cao et al. [34] found that inorganic carbon dissolved by carbonate rock outcrops was equal to 18% of the net carbon sink of terrestrial vegetation and 38% of the soil carbon sink. However, soil water and nutrient elements around rock outcrops in karst areas have been widely studied, but the role of soil water and carbon sinks around rock outcrops under continuous drought is still unknown. Therefore, this study illustrates the positive effects of rock outcrops on ecosystem water–carbon sinks and water–carbon coupling based on the rock–soil–water relationship.

To this end, this study simulated rock outcrops (RO) and non-outcrops (NRO) with different degrees of drought through indoor pot experiments, and the heterogeneous distribution characteristics of soil moisture and $HCO_3^-$ concentrations under drought in the two habitats were investigated. We aimed to evaluate the water retention capacity and the storage capacity of inorganic carbon in the two habitats and explore the relationship between soil water and $HCO_3^-$. The results will help to reveal the temporal variability patterns of soil water and inorganic carbon under drought conditions, provide a reference for the matching of heterogeneous habitats with suitable plants and, finally, improve the ecological restoration efficiency in karst areas.

## 2. Materials and Methods

### 2.1. Test Materials and Design

This experiment was carried out in a greenhouse of Guizhou Agricultural Vocational College, Guizhou province, China (26°34′ N, 106°25′ E). A constant light intensity of $500 \pm 50$ μmol·m$^{-2}$·s$^{-2}$ for 12 h per day was maintained by using LED lamps of type T5, with a day/night temperature of 25 °C/18 °C and relative air humidity of $55 \pm 5$%. The loamy soil was collected from the 0–20 cm cultivated layer of calcareous arable land with typical carbonate rock development in Qing Zhen City. The basic physicochemical properties are shown in Table 1. To simulate rock outcrops, containers measuring 50 cm, 21 cm, and 18 cm in length, width, and height were selected, with small holes at the bottom for gravity water to flow away. The soil was air-dried and milled until the particles were small and uniform, and the same mass (10 kg) of soil was weighed and poured into the containers.

**Table 1.** Basic physical and chemical properties of the soil.

| Soil Type | Particle Size/% | | | Nutrient Content/(g/kg) | | | | pH |
|---|---|---|---|---|---|---|---|---|
| | <0.002 mm | 0.02~0.002 mm | 0.02~2 mm | C | N | Mg | Ca | |
| Loam | 7.62 | 39.97 | 51.81 | 26.95 | 2.10 | 1.49 | 1.48 | 7.63 |

In contrast to non-karst areas, karst areas are characterized by the double space structure of the surface and underground (Figure 1a) [35,36]. At the same time, according to the field survey shown in Figure 1b, the bedrock of the outcrop habitat is exposed and has a low land use availability. Therefore, this study placed 1 kg of small limestone clasts at the bottom of the container to simulate the karst underground structure while, following the characteristics of the karst surface outcrops, some limestone was placed on one side of the container to simulate rock outcrops, and the pots were saturated with water, as shown in Figure 1c. In order to restore the disturbed soil to its original state, the container was placed outside for five months, and then a simulated rainfall experiment was conducted.

In line with the rainfall intensity standards from the literature and those adopted by China's meteorological data department, combined with the results of intense rainfall in the short calendar period of the past 30 years in southwest China [37], the simulated rainfall treatment was set up to be a high rainfall intensity of 70 mm/h. The soil hydrological characteristics in the two habitats and the diurnal and day-to-day variations in $HCO_3^-$ were observed under drought conditions. The diurnal variation was collected at the beginning

and every 2 h after the simulated rainfall, recorded as 0, 2, 4, 6, 8, 10, and 12 (H), respectively, as well as every 2 d after simulated rainfall, recorded as 0, 2, 4, 6, 8, 10, 12, 14, and 16 (D).

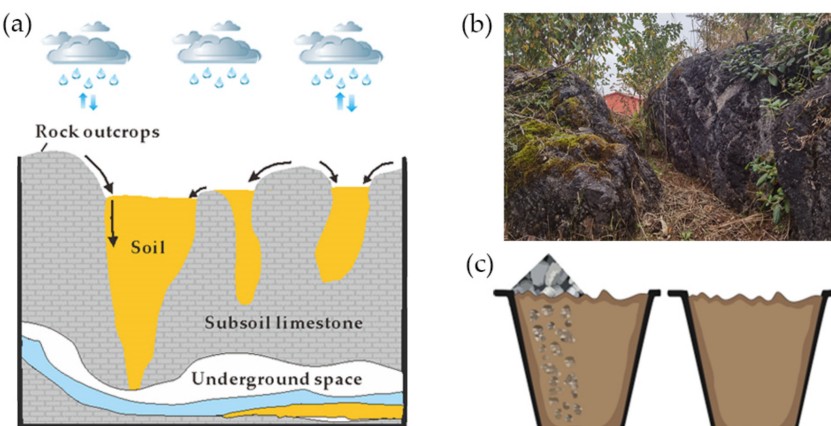

**Figure 1.** Illustration of karst rock outcrops. (**a**) Profile of soil water loss in rock outcrops. (**b**) Field karst rock outcrops. (**c**) Experimental setup.

### 2.2. Measurement of Soil Water Content, Water Potential, and Electrical Conductivity

TEROS 11/12 ($\theta$/EC) and TEROS 21 ($\varphi_S$) sensors connected with a ZL6 water potential meter (METER, Decagon, USA) were installed in two habitats to collect the dynamic soil water content (SWC), and water potential ($\varphi_S$) at a frequency of 10 min. The electrical conductivity (EC) was synchronously recorded.

### 2.3. Measurement of the Soil Surface Evaporation and Soil Infiltration

Evaporation from the soil surface (Ess) was determined by a continuous weighing method. The change in the soil water content in different habitats during the treatment period was calculated. The containers were weighed at 09:00 each day using an electronic scale. The difference in the evaporation container's weight, measured daily, was used as soil surface evaporation (Ems). Ems (kg/day) was converted to soil surface evaporation Ess (mm/day), which was calculated as follows [38]:

$$\text{Ess} = \text{Ems} \times 1000 \left(\text{cm}^3/\text{kg}\right) \times 10 \ (\text{mm/cm})/\text{A} \tag{3}$$

where A is the soil surface area ($\text{cm}^2$).

Soil infiltration was determined by using a graduated cylinder, water was collected every 10 min during the simulated rainfall (09:00–10:00) and every 2 h after the simulated rainfall (10:00–22:00).

### 2.4. Measurement of Soil Water Characteristic Curve

The soil water characteristic curve, an important indicator of the basic hydraulic properties of soil, describes the relationship between soil water content and soil water potential. It is usually used to estimate the soil water retention function and provides an accurate estimation method for soil hydrology [39]. By comparing various models, it has been found that the soil water characteristic curve fitted by the Gibbs free energy equation was more accurate [40], so this model was used in this study to fit the water characteristic curve; this model can be expressed as:

$$(1 + \text{P}) \ \varphi_S = \varphi_0 + k\ln(1 + \text{P}) \tag{4}$$

where $\varphi_S$ is the soil water potential (MPa), P is the soil water content (%), and $\varphi_0$ and k are model parameters. When P = 0, $\varphi_S = \varphi_0$, $\varphi_0$ represents the value of the soil water potential when the soil water content is 0, which reflects the adsorption capacity of soil to water.

### 2.5. Measurement of Soil Bicarbonate Concentration

Determination of soil $HCO_3^-$ concentration [41,42] was performed as follows: Firstly, a fresh soil sample was taken out of refrigeration, and 50 g soil matrix was accurately weighed and placed in a 500 mL beaker. According to a water–soil ratio of 2:1, 100 g deionized water was added into the beaker, and $CO_2$ in the water was removed after boiling and cooling. The sample was stirred and well mixed, then left to stand for 30 min before centrifuging ($5000\times g$, 30 min). The supernatant was transferred to a suction filtration bottle and passed through a 0.45 μm aqueous filter membrane, and the filtrate was used to determine the $HCO_3^-$ concentration (using an Alkalinity Determination Kit, Merck KGaA, Darmstadt, Germany). The pH was determined using the remaining filtrate in the beaker.

### 2.6. Statistical Analysis

The soil water characteristics and $HCO_3^-$ concentrations at different drought times in the two habitats were statistically analyzed. Differences between values for each treatment were analyzed with Duncan's multiple range tests combined with one-way ANOVA using SPSS software (SPSS 25.0, IBM, Armonk, New York, NY, USA). Drawings were made using Origin software (2021, Northampton, MA, USA). All experimental data are shown as means ± standard error, *n* = 3.

## 3. Results

### 3.1. Diurnal Variation in Soil Infiltration and Evaporation of the Two Habitats under Drought

Diurnal variations in soil evaporation and infiltration can be used to quantify the soil and water conservation effect of different habitats. Clear changes of the soil surface evaporation (Ess) and soil infiltration (SI) in the two habitats were observed, which were attributed to the continuous loss of surface soil water caused by rainwater infiltration and evaporation (Figure 2). Infiltrations in the two habitats showed similar changes after simulated rainfall (Figure 2a). However, infiltration in the non-outcrops was slightly higher than that in the rock outcrops at 11:00, and both infiltrations became stable and gradually fell to zero after 14:00. Figure 2b depicts the relationship between soil evaporation and time. The average evaporation of rock outcrops (4.15 mm/h) was significantly lower than that of non-outcrops (8.99 mm/h). Evaporation from the non-outcrops was the highest at 09:00 with a maximum evaporation of 38.87 mm/h, and the minimum was 1.24 mm/h, while the maximum evaporation from the rock outcrops was 17.30 mm/h and the minimum was 1.18 mm/h. Under drought conditions, soil evaporation in the non-outcrops had a greater fluctuation than that in rock outcrops, and both trends were first faster, then slower. Thus, soil infiltration in non-outcrops was significantly higher than that in rock outcrops, while soil evaporation in rock outcrops was much less than that in non-outcrops, and the magnitude of soil evaporation in rock outcrops decreased as time increased.

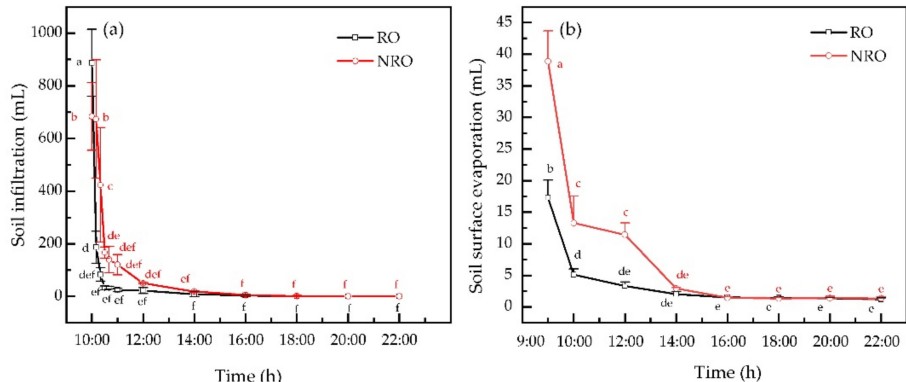

**Figure 2.** Diurnal variation in soil evaporation and infiltration in two habitats. (**a**) Changes in soil infiltration over time in two habitats. (**b**) Changes in soil evaporation over time in two habitats. RO and NRO represent rock outcrops and non-outcrops, respectively.

### 3.2. Diurnal and Day-to-Day Variations Soil Water Content and Water Potential in the Two Habitats under Drought

The trends of soil water content and water potential in the two habitats were basically the same (Figure 3). At the stage of diurnal variation (Figure 3a), the soil water content of rock outcrops was slightly higher than that of non-outcrops within 2 h after simulated rainfall, but there was no clear difference ($p > 0.05$) in the soil water content between the two habitats after 2 h (Figure 3a), and the mean values of the residual water content in rock outcrops and non-outcrops were 0.376 m$^3$/m$^3$ and 0.371 m$^3$/m$^3$, respectively. There was also no significant difference ($p > 0.05$) in the soil water potential of the habitats under the same soil water content conditions. Under diurnal variation, when the soil water potential in both habitats became stable, the water potential values ranged from $-8.933$ to $-9.132$ kPa in the rock outcrops and $-9.399$ to $-9.598$ kPa in the non-outcrops, respectively. In the first 10 consecutive days of drought, soil water content was slightly higher in the rock outcrops than in the non-outcrops, but the difference was not significant. After 10 days, soil water content was higher in the rock outcrops than that in the non-outcrops (Figure 3b). Furthermore, the water potential in both habitats showed gradual decreases with continuous drought, and a significant difference between 10 and 12 days was observed. Overall, the results indicated that soil water uptake was higher in non-outcrops than that in rock outcrops under severe water deficit, and the soil in non-outcrops was drier than that in rock outcrops. Under diurnal variation, the soil water content and water potential in both habitats were clearly influenced by the duration of drought.

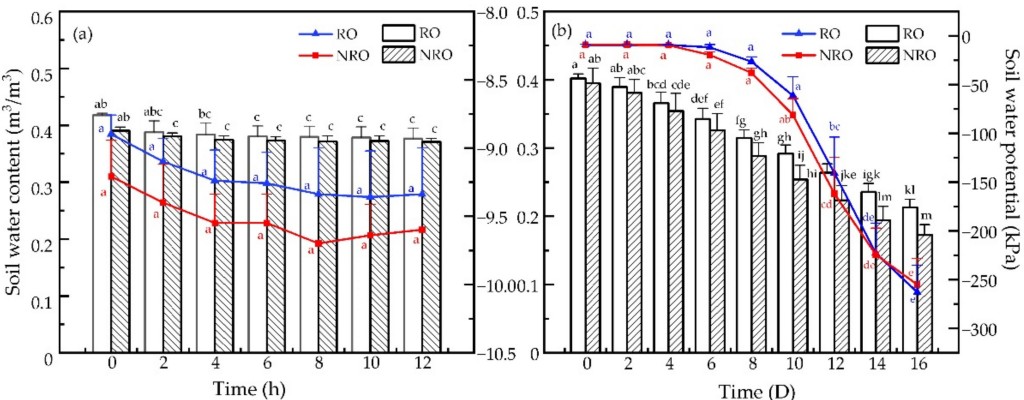

**Figure 3.** Diurnal and day-to-day variations in the soil water content and water potential in two habitats. (**a**) Diurnal variation of soil water content and soil water potential under drought conditions. (**b**) Day-to-day variation of soil water content and soil water potential under drought conditions. The bar graph and line graph represent the soil water content and soil water potential, respectively. Small letters indicate significant differences at 5% level $p < 0.05$ (Tukey).

### 3.3. Diurnal and Day-to-Day Variations in the Water Characteristic Curves of the Two Habitats under Drought

Soil water potential and water content were linearly fitted by the Gibbs free energy equation to investigate the soil water characteristic curves in both habitats (Table 2). This study found that the correlation coefficients between soil water potential and water content under each drought level were all above 0.97, showing a significant positive correlation. The parameter $y_0$ and K values in the water characteristic curve equation represent the water sorption capacity of soil and the water retention function of soil, respectively [40], while the $y_0$ value also characterizes the decrease speed of soil water content with decreasing soil water potential [43]. The K value in the diurnal variation was significantly greater in the rock outcrops (0.0245) than in the non-outcrops (0.0134), while the $y_0$ value in the non-outcrops ($-0.0127$) was greater than in the rock outcrops ($-0.0159$). Therefore, the soil water-holding capacity of the rock outcrops was significantly higher than that of the non-outcrops, and the rate of change of soil water content and water potential in the rock

outcrops was lower than that of the non-outcrops. There was also a similar trend in the day-to-day variation: the K value of the rock outcrops (5.0316) was significantly greater than that of the non-outcrops (3.3624), and the $y_0$ value of the non-outcrops (−0.6819) was greater than that of the rocky habitat (−1.0396). In summary, the water sorption capacity of rock outcrops was higher than that of non-outcrops under both diurnal and day-to-day variations, and the soil water content of non-outcrops decreased more rapidly with decreasing soil water potential than that of rock outcrops. Therefore, the overall soil water holding capacity of rock outcrops was higher than that of non-outcrops.

**Table 2.** Fitting effects and equations of soil water characteristic curves in two habitats.

| Habitat Type | Equations and Parameters | | | | |
| --- | --- | --- | --- | --- | --- |
| | $R^2$ | K | $y_0$ | $p$ | Equation |
| RO | 0.9943 | 0.0245 | −0.0159 | <0.0001 | $\varphi_S = [-0.0159 + 0.0245\ln(1 + P)/(1 + P)]$ |
| NRO | 0.9730 | 0.0134 | −0.0127 | <0.0001 | $\varphi_S = [-0.0127 + 0.01341\ln(1 + P)/(1 + P)]$ |
| $RO_C$ | 0.9815 | 5.0316 | −1.0396 | <0.0001 | $\varphi_S = [-1.0396 + 5.0316\ln(1 + P)/(1 + P)]$ |
| $NRO_C$ | 0.9906 | 3.3624 | −0.6819 | <0.0001 | $\varphi_S = [-0.6819 + 3.3624\ln(1 + P)/(1 + P)]$ |

Note: $RO_C$ and $NRO_C$ represent rock outcrops and non-outcrops, respectively, during day-to-day variations stage. $p$ indicates a significance of 0.05; $\varphi_S$ is the soil water potential; $p$ is the soil water content.

### 3.4. Diurnal and Day-to-Day Variations in Soil Conductivity, $HCO_3^-$, and pH in Two Habitats under Drought

3.4.1. Dynamic Change in Soil Conductivity with Increasing Drought Time

Figure 4 shows the variation patterns of soil conductivity with increasing drought time in the two habitats. According to the standards of soil salinization grade classification [44], the two habitats belong to non-salinization soil. The soil conductivity under diurnal variation in the rock outcrops was at a maximum at 0 h from the onset of the simulated rainfall, and then decreased to its lowest at 2 h, and gradually increased over time from 4–12 h. The variation in soil conductivity in the non-outcrops was not significant (Figure 4a). Under day-to-day drought conditions, soil conductivity increased sharply at the beginning of the rainfall and then gradually decreased (Figure 4b). There was a significant difference in soil conductivity between the two habitats ($p < 0.05$), the soil conductivity of rock outcrops was significantly higher than that of non-outcrops, and the difference between the two habitats gradually decreased with the continuing drought. Soil conductivity in the rock outcrops did not change significantly from 0 to 6 d, and it decreased continuously with increasing drought time after 6 d, while the non-outcrops showed a slow decreasing trend with drought.

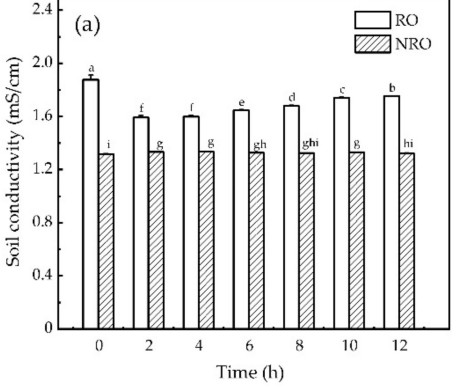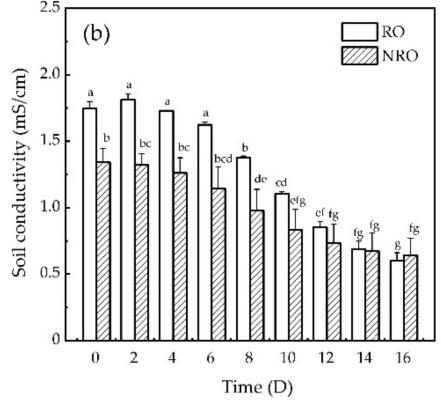

**Figure 4.** Diurnal and day-to-day variations in soil conductivity in two habitats. (**a**) Diurnal variation of soil conductivity under drought conditions. (**b**) Day-to-day variation of soil conductivity under drought conditions. RO and NRO represent the rock outcrop habitat and the non-outcrop habitat, respectively. Small letters indicate significant differences at 5% level $p < 0.05$ (Tukey).

### 3.4.2. Dynamic Changes in Soil pH and $HCO_3^-$ with Drought Time

Soil $HCO_3^-$ concentration and pH varied significantly with drought time. Under diurnal variation (Figure 5a), soil $HCO_3^-$ concentrations in both rock outcrops and non-outcrops reached the maximum values (4.8042 mmol/L and 3.5373 mmol/L) at 10 h after simulated rainfall. The soil solution pH changed slightly in the non-outcrops, while no clear change was observed in the rock outcrops. Under day-to-day drought conditions (Figure 5b), the soil $HCO_3^-$ concentration and pH varied remarkably as time increased, with significantly higher $HCO_3^-$ concentration and pH in the rock outcrops than in non-outcrops, especially in the first 4 d of continuous drought, when both the $HCO_3^-$ concentration and pH in rock outcrops reached a peak of 3.9909 mmol/L and 7.76 mmol/L, respectively. After 4 to 6 d of drought, the $HCO_3^-$ produced by dissolution dissociated into $CO_2$ and $H_2O$, and the $CO_2$ escaped, after which it showed a stable state. Therefore, in the drought period, the dynamic changes of $HCO_3^-$ in the rock outcrops habitat with pH first increased, then degassed, and finally stabilized, achieving a zero-carbon sink. In contrast, the soil $HCO_3^-$ concentration in non-outcrops did not change remarkably throughout the drought period, and its pH value increased incrementally after simulated rainfall then began to decrease, and slightly increased at the end the treatment. Figure 5a shows that the soil reached a high saturation level after simulated rainfall, and the $H^+$ in the soil gradually diluted with increasing water content, resulting in a gradual increase in pH and $HCO_3^-$ content. A similar trend was observed under day-to-day variations. In the first 4 d after simulated rainfall, the $HCO_3^-$ in soil was enriched with increasing drought time because the soil water content was still in a more saturated state; after 4 consecutive d of drought, the $HCO_3^-$ content and pH in the soil started to show a decreasing trend.

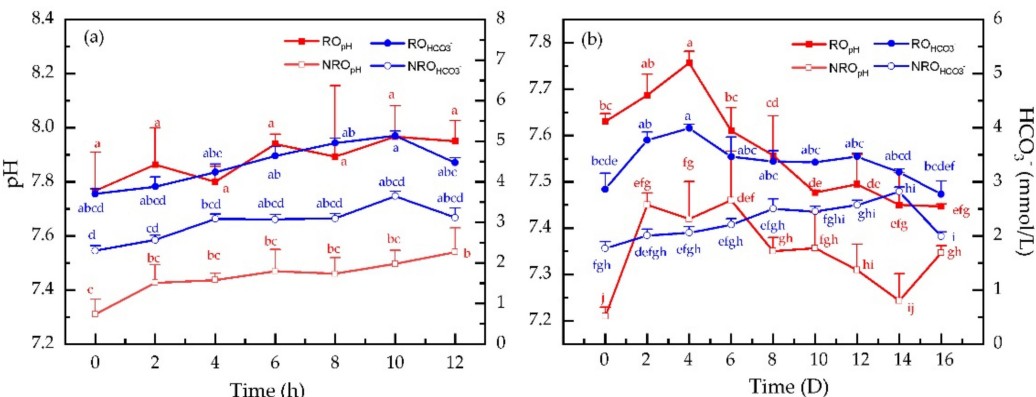

**Figure 5.** Diurnal and day-to-day variations in soil $HCO_3^-$ and pH in two habitats. (**a**) Diurnal variation of soil $HCO_3^-$ and pH under drought conditions. (**b**) Day-to-day variation of soil $HCO_3^-$ and pH under drought conditions. RO and NRO represent the rock outcrop habitat and the non-outcrop habitat, respectively. Small letters indicate significant differences at 5% level $p < 0.05$ (Tukey).

### 3.4.3. Correlation of Soil Parameters in Two Habitats

In this study, the relationships between EC, pH, $HCO_3^-$, SWC, and $\varphi_S$ in two habitats was investigated. In rock outcrops (Figure 6a), EC was strongly positively correlated with pH, SWC and $\varphi_S$ ($p < 0.01$). pH showed a strong positive correlation with SWC ($p < 0.01$), and a positive correlation with $\varphi_S$ ($p < 0.05$). SWC was strongly positively correlated with $\varphi_S$ ($p < 0.01$). In non-outcrops (Figure 6b), EC was strongly positively correlated with SWC and $\varphi_S$ ($p < 0.01$), and a negatively correlated with $HCO_3^-$ ($p < 0.05$). SWC was strongly positively correlated with $\varphi_S$ ($p < 0.01$). The relationship between soil $HCO_3^-$ concentration and other parameters in rock outcrop habitats during drought was not significant ($p > 0.05$). However, soil $HCO_3^-$ concentrations in non-outcrop habitats during drought had a significant negative correlation only with EC, but not with any other parameters ($p > 0.05$).

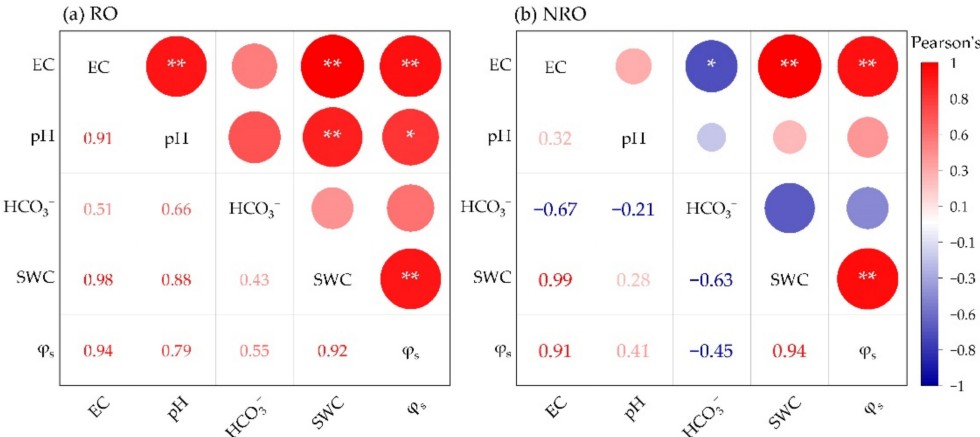

**Figure 6.** Pearson correlation coefficients between different parameters in soils of the two habitats. (**a**) indicates Pearson correlation analysis in rock outcrops habitat. (**b**) indicates Pearson correlation analysis in non-outcrops habitat. RO and NRO represent the rock outcrops habitat and the non-outcrops habitat, respectively. ** Correlation is significant at the 0.01 level (2-tailed). * Correlation is significant at the 0.05 level (2-tailed).

## 4. Discussion

### 4.1. Water Retention Function of Soils under Drought in the Two Habitats

Karst soil water is influenced by a combination of rainfall, surface rock exposure, drought time, and geological structure, showing strong spatial and temporal heterogeneity [45]. It was found that the average evaporation of rock outcrops (4.15 mm/h) was significantly lower than that of non-outcrops (8.99 mm/h) in daily variation. At the same time, the soil water content of rock outcrops was significantly higher than that of non-outcrops on d 10 from the onset of drought. Rock outcrops have positive hydrological effects during drought periods, and different drought times and levels not only affect soil water content, but also lead to redistribution of soil water to a more uniform state, horizontally and vertically [42]. Our study showed that the rock outcrops in a karst area had a significant positive effect on soil water accumulation. With a continuous increase in drought time, the redistribution of water generated by runoff processes creates a heterogeneous spatial distribution of soil water [46], and the water retention function of soils in rock outcrops increased significantly. Under diurnal drought, rock outcrops are considered a potential factor influencing soil water distribution near exposed rocks, especially after rainfall [47]. Our results showed that the soil water content of the two habitats had a downward trend only for 0–4 h, and then changed steadily at the rates of 0.2% and 0.16%, with mean values of 0.381 m³/m³ and 0.374 m³/m³ respectively. The soil water potential changed slightly during the day (Figure 3). However, under day-to-day drought conditions, this phenomenon was very obvious (Figure 3). Both reduced evapotranspiration due to rock shading and infiltration of runoff at the rock–soil interface are expected to increase soil moisture near rock outcrops [11]. Within the drought period, non-outcrops showed a quick soil water response to rainfall and more pronounced evaporation and infiltration changes, while the rock outcrops created a heterogeneous spatial distribution of soil water during rainfall, which was attributed to the redistribution of rainfall caused by rock surface runoff [48,49], making the soil water content in rock outcrops relatively stable during continuous drought. The soil water content around the rock outcrops was significantly higher than that of the non-outcrops, indicating that the water retention function of the soil in the rock outcrops was greater than that of the non-outcrops.

### 4.2. Carbon Sequestration and Sinks in Soils of the Two Habitats under Drought

Soils in karst regions are often affected by drought, but rainfall in the region is actually quite abundant. The solubility of surface carbonate rocks depends on the pH of the water around it [50,51]; abundant rainfall promotes the dissolution of carbonate rocks and the

production of soil $CO_2$ [52,53], which is also an important driver of carbon flux. It was observed that the presence of surface limestone promoted the increase of soil $HCO_3^-$ concentration in rock–soil–water interactions. The average concentration of $HCO_3^-$ in the soil of rock outcrops was 3.36 mmol/L, while that of non-outcrops was 2.26 mmol/L. The soil $HCO_3^-$ concentration was significantly different among different habitats, and the soil $HCO_3^-$ concentration in rock outcrops was higher than that in non-outcrops. A portion of the absorbed carbon could be stored in the soil solid phase, when the soil has a high pH and sufficient calcium [54]. Results of this study indicated that the soil $HCO_3^-$ concentration in rock outcrops fluctuated with the dynamics of drought and rainfall. In the early stages of drought, soil patches in rock outcrops habitats have sufficient water content and the soil $CO_2$ is consumed by carbonate dissolution [55], which led to the increase of soil pH and $HCO_3^-$ concentration generated by dissolution at the rate of 37.6% and 9.6%, respectively, within 0–4 d of drought. Although there was no significant difference in soil conductivity values, the values also showed a slightly increasing trend. Furthermore, soil pH and salinity affect carbonate dissolution and precipitation through biochemical and physico–chemical processes during the carbon cycle [54]. Therefore, the carbon sink in the soil of the rock outcrops habitat is increased relative to the non-outcrops, which is related to the soil water content, increased $CO_2$ concentration, and $CaCO_3$ dissolution [55]. At present, many studies have been carried out on the link between pH and carbon. Varadharajan et al. and Saaltink et al. have reported that the pH affects the carbon balance in the soil [56,57]. Meanwhile, Drysdale et al. and Montety et al. also revealed that temperature changes could affect pH, and hence the solubility of carbonate rocks, increasing $CO_2$ escape [58,59]. This was also reflected in our study, where pH in rock outcrops gradually decreased from d 4 of drought onwards, and their soil conductivity and $HCO_3^-$ concentrations also decreased. On d 0 to 4 from the onset of the drought, the soil $HCO_3^-$ concentration gradually increased and a carbon sink occurred. Subsequently, from d 4 to 6, the soil $HCO_3^-$ concentration began to decrease, the $CO_2$ escaped, and soil degassing gradually appeared. Some studies have found that soil $HCO_3^-$ can promote plant growth [60] and be used as a carbon source for plant uptake [61]. Therefore, rock outcrops have more carbon resources than non-outcrops, which can promote photosynthesis of plants. However, the carbon uptake and release of rock outcrops tended to be stable throughout the drought period, and the carbon sink gradually became zero. Our results are consistent with basic ecological theory, that is, the carbon sink of a given ecosystem will inevitably approach zero in an undisturbed natural state, which is attributed to of the long-term dynamics of the ecosystem [62]. In comparison, the variation in soil $HCO_3^-$ concentration in non-outcrops during the drought period was not significant, thereby also indirectly corresponding to the possibility that the carbon sinks occurring in rock outcrops are only from rocky dissolution, and the depletion of atmospheric/soil $CO_2$ is not significant.

### 4.3. Coupling Effect of Soil Water Content and $HCO_3^-$ under Drought

Our research found that $HCO_3^-$ concentrations in soils of rock outcrops were not significantly related to SWC throughout the drought period, whereas $HCO_3^-$ in non-outcrops was negatively correlated with EC (Figure 6). In the study by Yang et al. [63], soils with higher water content and lower temperature were reported to promote $CO_2$ fixation, while soil $CO_2$ was gradually consumed by carbonate dissolution, increasing the dissolved carbonate content [64]. Cao et al. found that the lower the rainfall and temperature, the lower the carbonate rock erosion rate and carbon sink became [65]. This is consistent with the results in our study, where $HCO_3^-$ in the soil showed large differences between drought times. The dynamic changes in soil $HCO_3^-$ concentration with declining water content in the two habitats can be divided into two stages: enrichment and degassing. The two stages in the rock outcrops were an initial enrichment stage of 0–4 d and a degassing stage of 4–16 d; the two stages in the non-outcrops were an initial enrichment stage of 0–14 d and a degassing stage of 14–16 d. In the first stage, the soil of the rock outcrops had slight evaporation, insignificant changes in soil conductivity, and a high water retention capacity,

and this gave the retained water in the soils of rock outcrops sufficient reaction time to dissolve $CO_2$ as well as to promote the dissolution of limestone. With increasing time, the soil solution pH and $HCO_3^-$ concentration increased more rapidly due to the influence of $CO_2$ and limestone dissolution, consuming more soil $CO_2$ and continuously enriching the $HCO_3^-$ in the soil patches. However, in the second stage, the soil water content decreased significantly, which limited the ability of soil to obtain the atmospheric $CO_2$ for carbon assimilation and weakened the process of limestone dissolution. The combination of $H^+$ and $HCO_3^-$ in the soil released $CO_2$ into the soil and atmosphere, leading to the increased concentration of free $CO_2$ in the water column and a significant decrease in the soil solution pH and soil conductivity; this was the time at which the soil degassing effect gradually appeared. While the rate of increase of the soil $HCO_3^-$ concentration in the non-outcrops was not obvious in the first stage. In the second stage, under continuous drought, the soil water content decreased significantly, leading to a gradual decline in the soil $HCO_3^-$ concentration and the beginning of degassing in non-outcrops. Moreover, the trends of pH and conductivity in non-outcrops were consistent with those in rock outcrops, and their values were lower than those in rock outcrops. In conclusion, under water-saturated conditions, soil $HCO_3^-$ concentration increased gradually and was enriched, Under day-to-day drought conditions, degassing of $HCO_3^-$ occurs. Soil $HCO_3^-$ fluctuated with changes in the water content, thereby causing changes in the carbonate dissolution environment.

### 4.4. Limitations of the Study

The present study has several limitations. Firstly, the indoor potted simulated outcrop habitat is small and single in type, and the geological complexity of karst areas and the complex and variable patterns of soil water infiltration around rock outcrops are still unknown and need further research. Secondly, our experiments were carried out under ideal conditions, whereas rock outcrops in the field can be subject to the combined effects of weather, flora and fauna, and outcrop extent at any time. Consequently, our indoor simulation results may differ from those of field runoff infiltration and rock outcrop inorganic carbon dynamics. Finally, the current experimental results are limited to describing the dynamics of soil water and inorganic carbon in rock outcrops in stable ecological environments. Therefore, our study only provides the basis and rationale for further field experiments in rock outcrops.

### 5. Conclusions

This study quantitatively investigated the temporal variability of soil water carbon in rock outcrops and non-outcrop habitats under drought conditions, and discussed the response mechanism of soil water carbon in the two habitats to drought. We found that the evaporation of rock outcrops was slower and the soil water retention capacity was stronger than that of non-outcrops. Moreover, rain promoted the dissolution of limestone, which resulted in 1.49 times higher carbon sequestration capacity in the rock outcrops than in the non-outcrops. Furthermore, the inorganic carbon content of soils in karst areas was closely related to the soil water content, and other environmental factors. During the water deficit stage, a close to saturation water content in rock outcrops promoted carbonate dissolution, which changed the $HCO_3^-$ concentrations in the soil solution from 2.862 mm/L to 3.991 mm/L, while low soil water content caused soil $HCO_3^-$ to be released into the soil or atmosphere as $CO_2$. As a result, the $HCO_3^-$ concentrations in the soil solution changed from 3.991 mm/L to 2.772 mm/L. Therefore, the soil $HCO_3^-$ concentration fluctuated with the external environmental conditions. In the rock–soil–water interaction, the soil $HCO_3^-$ concentration in the rock outcrops in the early stage of drought caused a transient pulse phenomenon, but the overall carbon balance tended to be stable, showing the phenomenon of zero-carbon sinks. The variation in soil $HCO_3^-$ concentration in non-outcrops was not significant during the drought period, which indirectly indicated that the carbon sink in rock outcrops was mainly derived from the dissolution of rocks, and the depletion of atmospheric/soil $CO_2$ was not significant. Therefore, this study of

the temporal characteristics of soil water and carbon under drought conditions provides a reference for matching the best crops to fragile karst ecosystems.

**Author Contributions:** Y.W. constructed and conceived the project. Y.W. and L.L. designed the research. L.L., Y.Z. and H.L. performed the research. L.L., A.X. and D.X. analyzed the data. L.L., D.X. and Y.W. wrote the paper. All authors have read and agreed to the published version of the manuscript.

**Funding:** This research was funded by Support Plan Projects of Science and Technology of Guizhou Province [number (2021) YB453].

**Data Availability Statement:** The data presented in this study are available on request from the corresponding author.

**Acknowledgments:** We are thankful to H.L. for providing the laboratory platform and experimental facilities, and to D.X. for his technical support during the experiments.

**Conflicts of Interest:** The authors declare no conflict of interest.

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
