# Peer review of "Drought Induced Dynamic Traits of Soil Water and Inorganic Carbon in Different Karst Habitats"

_water, doi:10.3390/w14233837_

Round 1
Reviewer 1 Report (Previous Reviewer 2)
Very minor corrections are needed (see corrected manuscript v2)

Author Response
Reviewer #1:
Comment: Very minor corrections are needed (see corrected manuscript v2).
Response: Thank you very much for your advice, we have revised all according to “corrected manuscript v2”.
We hope we can obtain your approval of the revised manuscript. Thank you very much for the time and effort you have spent on this paper.
Reviewer 2 Report (Previous Reviewer 3)
Dear Authors,
after reading the revised versione of your manuscript I found it improved and now ready for publication after two minor corrections:
1) Caption of Fig. 2 is not in the right place
2) In the title of the sub-chapter "3.4.3. Correlation of Soil Hydrological Parameters in Two Habitats" the word "hydrological" should be deleted, because you discuss pH, bicarbonate concentrations and electric conductivity, which are chemico-physical and not hydrological parameters.
Author Response
Reviewer #2
Comment 1: Caption of Fig. 2 is not in the right place.
Response: Thank you for your suggestion. The position of Figure 2 has been modified.
Comment 2: In the title of the sub-chapter “3.4.3. Correlation of Soil Hydrological Parameters in Two Habitats" the word "hydrological" should be deleted, because you discuss pH, bicarbonate concentrations and electric conductivity, which are chemico-physical and not hydrological parameters.
Response: Thank you for your suggestion. The title of sub-chapter 3.4.3 has been changed as required.
We hope we can obtain your approval of the revised manuscript. Thank you very much for the time and effort you have spent on this paper.
Reviewer 3 Report (Previous Reviewer 5)
The author has carefully revised, but the language should be further polished.
Author Response
Reviewer #3:
Comment: The author has carefully revised, but the language should be further polished.
Response: We thank you for suggesting to polish the language. We have polished our manuscript carefully and corrected the grammatical, styling, and typos found in our manuscript.
We hope we can obtain your approval of the revised manuscript. Thank you very much for the time and effort you have spent on this paper.
This manuscript is a resubmission of an earlier submission. The following is a list of the peer review reports and author responses from that submission.
Round 1
Reviewer 1 Report
It is a good work but few clarification are required
-Ro and NRO field details required , some field photos may be required
-the earlier work based on field observations and field experiments need to be added ( earlier literature )
-discussion part needs to add more data driven discussion rather than speculation
-conclusion should have more data driven
Reviewer 2 Report
The presentation of the problem, the method and the findings is clear. Some additional references could be made to field studies in this topic. The rainfall intensity used in the experiment is presented as medium value but I find it too high. Plese, check and explain.
I made some suggestions to make the English formulation easier to understand (see attached file).

Reviewer 3 Report
Dear Authors,
I tried to evaluate your manuscript and started to give some suggestions, as in the annotated attached pdf.
I had to stop my revision because the English style, and the construction of the most of the sentences, are so poor that it is impossible to clearly understand what you mean.
Your submission needs a very deep linguistic revision prior to be sent to a peer review process.
Moreover, there is a disproportion between Chinese (44 titles) and not-Chinese (16) authors cited in your reference list: please revise the reference list making it more international.

Reviewer 4 Report
The manuscript is well written in the relevant area. It is clear, concise and comprehensive. It is interesting to the readers of water. However, before accepting this manuscript for publication, I request the authors to address the following comments/concerns during revision.
1. The introduction is not clearly defined. Kindly add some relevant recent studies.
2. English usage must be improved for better clarity
3. The material and methods section need improvement.
4. How this study is much better than other available methods.
5. Kindly concise the conclusion and make it relevant to the readers.
What are the policy recommendations at national and at global level?
Reviewer 5 Report
1. In the study, it is said that there are two kinds of habitats, but in fact, there is only one contrast reference without rocks. As the result, "the rock outcrops had less evaporation and significantly greater water retention capacity than the non outcrops", according to common sense, under the same evaporation area, the evaporation area with bedrock is certainly smaller than that without bedrock. What is the origin of the research "habitat"?
2. The “2. Materials and Methods” are not well understood. For example, how to understand "According to the unique structure of the karst": and what is the basis for the placement of rocks and soil? I don't understand the monitoring in the field for up to 5 months. Then simulate rainfall under different drought conditions? In short, it is very confusing.
3. Pot experiment scale is too small, the author how to avoid experimental error ?
4. n=3 refers to the repetition of various indicators in the study? There are only two test devices? Are all the samples from these two devices?
5. It is recommended that the author supplement the limitations of the study.
6. Fonts in the chart and figure are not uniform.
7. There are some errors in the references, such as the time in Reference 16.